# Stereodivergent dehydrative allylation of β-keto esters using a Ru/Pd synergistic catalyst

Thien Phuc Le[1], Shinji Tanaka [2]✉, Masahiro Yoshimura [3], Kazuhiko Sato [2] & Masato Kitamura[1]✉

α-Alkylation of a β-keto ester is a frequently used reaction for carbon–carbon bond formation. However, extension to a stereoselective reaction remains a significant challenge, because the product easily racemizes under acidic or basic conditions. Here, we report a hybrid system consisting of Pd and Ru complexes that catalyzes the asymmetric dehydrative condensation between cinnamyl-type allylic alcohols and β-keto esters. α-Non-substituted β-keto ester can be allylated to afford an α-mono-substituted product with high regio-, diastereo-, and enantioselectivity. No epimerization occurs owing to the nearly neutral conditions, which is achieved by a rapid proton transfer from Pd-enolate formation to Ru π-allyl complex formation. Four diastereomers can be synthesized on demand by changing the stereochemistry of the Pd or Ru complex. Eight stereoisomers with three adjacent stereogenic centers can be synthesized by employing diastereoselective reduction of the ketone in the products. The formal synthesis of (+)-pancratistatin demonstrates the utility of the reaction.

β-Hydroxy esters (βHEs) are ubiquitous structural motifs in organic synthesis, and the asymmetric hydrogenation of β-keto ester is the most promising for providing βHEs[1]. Although α-alkylated chiral βHE is frequently requested for constructing other complicated structures, the α-alkylation of βHEs is inefficient. Acetalization, deprotonation, alkylation, and deacetalization are generally required for diastereoselective synthesis[2]. Diastereoselective and enantioselective hydrogenation of α-substituted β-keto esters via dynamic kinetic resolution is attractive[3]. However, the substrate scope is limited, and the product is diastereospecific. From this perspective, the enantioselective alkylation of β-keto ester, also known as the acetoacetic ester synthesis, seems alluring, followed by diastereoselective reduction of the carbonyl group.

Acetoacetic ester synthesis[4], which was discovered by Geuther in 1863, is a traditional and basic carbon–carbon bond-forming reaction wherein the β-keto ester is alkylated via an enol or enolate formation and reaction with alkyl halides[5]. Because highly functionalized compounds can be synthesized from ample substrates, this reaction has

been widely used in organic syntheses. Accordingly, various enantioselective reactions have been developed[6]. However, there is a critical and fundamental limitation from the viewpoint of asymmetric synthesis. The enantioselective acetoacetate synthesis is generally applied to afford a quaternary carbon center using an α-alkylated β-keto ester substrate[4–6]. The tertiary stereogenic center is generated when a non-substituted β-keto ester is alkylated. However, the enantiopurity of the generated product is generally low because the corresponding product easily racemizes through keto–enol equilibrium, particularly under basic or acidic conditions. Although enolate formation by a base is vital to proceed the reaction, this makes it difficult to avoid racemization. Thus, the development of mono-alkylation of β-keto ester in an enantioselective manner has remained a significant issue[7].

To solve this problem, we focused on Tsuji–Trost-type (T–T) asymmetric allylation[8–12], which is a typical catalytic reaction for the alkylation of 1,3-dicarbonyl compounds. The T–T reaction can construct a chiral carbon center at two positions, the allylic carbon and dicarbonyl methylene carbon[13]. In the case of dicarbonyl methylene

[1]Graduate School of Pharmaceutical Sciences and Research Center for Materials Science, Nagoya University, Chikusa, Nagoya 464-8601, Japan. [2]Interdisciplinary Research Center for Catalytic Chemistry, National Institute of Advanced Industrial Science and Technology (AIST), Tsukuba 305-8565, Japan. [3]Division of Liberal Arts and Sciences, Aichi Gakuin University, Nisshin 470-0195, Japan. ✉e-mail: tanaka-sh@aist.go.jp; kitamura@ps.nagoya-u.ac.jp

**Fig. 1 | Enantioselective allylation of β-keto esters. a** Enantioselective allylation of α-monosubstituted β-keto ester. **b** Enantioselective allylation of an unsubstituted β-keto ester. **c** Stereodivergent allylation of β-keto ester (this work).

**Fig. 2 | Basic strategy for stereodivergent catalytic dehydrative allylation systems.** $M^1$ and $M^2$: metal atom of complex. $L^1$ and $L^2$: chiral ligand. X: anionic ligand.

carbon, the reaction can be applied only to quaternary carbon formation, as shown in Fig. 1a. Double stereo-differentiation occurred when the branched allylation proceeded, as illustrated in Fig. 1b. However, only the stereochemistry of the allylic position can be arranged in this strategy, generating a stereoisomeric mixture (Fig. 1b)[13] due to epimerization via keto–enol equilibrium[14–17]. Thus, the stereodivergent synthesis has not been realized to date. Herein, we attempted to construct both the stereogenic centers using a binary catalytic system, as shown in Fig. 1c[18–35]. The product possessed many transformative functionalities with two chiral centers in a small molecule. This method can be extended to the synthesis of complicated chiral organic compounds.

In this study, we establish the catalytic asymmetric dehydrative allylation of non-substituted β-keto esters. The catalyst design, screening result, substrate scope and limitation, consideration of reaction mechanism, extension to the construction of three adjacent stereocenters via diastereoselective reduction of ketone, and synthesis of (+)-pancratistatin are reported.

## Results and discussion

### Development of the catalytic system

A basial strategy is illustrated in Fig. 2, which is based on stereo-divergent allylation using binary chiral catalyst systems, $M^1XL^1$ and $M^2L^2$, where $L^1$ and $L^2$ are chiral ligands. Catalyst $M^1XL^1$ activated β-keto ester **1** to form a corresponding metal enolate ($M^1$ enolate). Here, the anionic X functioned as a Brønsted base to abstract the proton of β-keto ester **1** with the liberation of conjugate acid, HX. Catalyst $M^2L^2$ activated allylic alcohol **2** to form a π-allyl complex ($M^2$ π-allyl). $M^2L^2$ combined with the liberated HX can cleave a stable alcoholic C–O bond based on the Redox-mediated Donor–Acceptor Bifunctional Catalyst (RDACat) concept[36]. The formation of hydrogen bond between the allylic alcohol and HX activated the C–O bond, followed by the oxidative addition with $M^2L^2$, which was accompanied by the release of water. The reaction between nucleophilic $M^1$ enolate and electrophilic $M^2$ π-allyl produced the allylation adduct **3**. The amount of HX generated during the catalytic cycle was the same as that of the introduced catalyst. Thus, the reaction proceeded under almost neutral conditions to avoid epimerization. Each chiral circumstance of $M^1L^1$ and $M^2L^2$ could discriminate the enantioface selection of enolate and π-allyl moieties, respectively. Thus, the selection of $M^1XL^1$ and $M^2L^2$ is crucial. We focused on CpRu–(S,S)-Naph-diPIM-dioxo-*i*Pr

($RuL^2_S$) as $M^2L^2$, which catalyzes dehydrative asymmetric allylation of 1,3-dicarbonyl compounds combined with Brønsted acid[37,38]. However, the selection of $M^1XL^1$ was a vital issue.

*tert*-Butyl 3-oxopropanoate (**1a**, 500 mM) and cinnamyl alcohol (**2a**, 600 mM) were selected as standard substrates and investigated as $M^1XL^1$ under the following conditions: 2 mol% of $M^1XL^1$, 2 mol% of $RuL^2_S$, 1,4-dioxane, and 10 °C; the results are shown in Table 1. A complete consumption of **1a** within 6 h was observed when Pd((R)-BINAP)(H$_2$O)$_2$(OTf)$_2$ (Pd$L^1_R$(OTf)) was used[39,40], generating **3aa** with high diastereo- and enantioselectivity (Table 1, entry 1). Among the four stereoisomers, (R,S)-*syn*-**3aa** was obtained in 99.7% yield while yields of (S,S)-*anti*-**3aa**, (S,R)-*syn*-**3aa**, and (R,R)-*anti*-**3aa** were 0.1, 0.2, and <0.1%, respectively. Each catalyst, Pd$L^1_R$(OTf) and Ru$L^2_S$, could be reduced to 0.25 mol% without the loss of yield and selectivity by elongating the reaction time (entries 2 and 3). Although the combination of Ru$L^2_R$ and Pd$L^1_R$(OTf) (2 mol% each) afforded (R,R)-*anti*-**3aa** with high enantioselectivity, it demonstrated lower reactivity and diastereoselectivity (37%, 85:15 diastereomeric ratio (dr)); the reaction was completed in 24 h (entries 4 and 5). Consequently, the combination of Ru$L^2_R$ and Pd$L^1_R$(OTf) was a mismatched system, while that of Ru$L^2_S$ and Pd$L^1_R$(OTf) was a matched system. According to the investigation of conditions in a mismatched system, decreasing the amount of Pd$L^1_R$(OTf) to 1 mol% increased the diastereoselectivity, generating **3aa** with a dr of 96:4 in 98% yield after 72 h under lower concentrations (entries 6–8). In contrast to (R)-BINAP ($L^1_R$) in Pd$L^1_R$(OTf), the more sterically hindered tol-BINAP and xyl-BINAP exhibited less reactivity (entries 9 and 10). The Pd complex of (R)-SEGPHOS exhibited reactivity and selectivity similar to that of BINAP (entry 11) without changing the stereochemistry. The reaction proceeded with the exchange of counter anion of Pd$L^1_R$(OTf) with BF$_4$, although the reactivity and diastereoselectivity were less. Conversely, the reaction did not proceed in the case of OTs (entries 12 and 13). Pd was essential as $M^1$, while the corresponding Ni complex was ineffective (entry 14). In addition, the Cu complexes did not afford any desired product (entries 15 and 16). Compared to the previously reported allylation of Meldrum's acid, the Brønsted acid, TfOH, or *p*-TsOH, could not function as a co-catalyst[37], resulting in no reactivity (entries 17 and 18). Additionally, the reaction did not proceed without the Ru catalyst (entry 19). In both matched and mismatched catalyst systems, under higher temperatures, diastereoselectivity tended to become lower (further details are provided in the Supplementary Information).

### Substrate scope of the reaction

The generality of β-keto ester **1** using cinnamyl alcohol (**2a**) was investigated in the Pd$L^1_R$(OTf)/Ru$L^2_S$–matched catalyst system using

**Table 1 | Screening of two metal complexes catalyzing the dehydrative allylation of 1a and 2a**

| Entry | M¹XL¹(mol%) | RuL² (mol%) | Time (h) | Yield (%) | (R,S) | (S,S) | (S,R) | (R,R) |
|---|---|---|---|---|---|---|---|---|
| 1 | PdL¹$_R$(OTf) (2) | RuL²$_S$ (2) | 6 | >99 | 99.7 | 0.1 | 0.2 | <0.1 |
| 2 | PdL¹$_R$(OTf) (1) | RuL²$_S$ (1) | 12 | >99 | 99.5 | 0.4 | 0.1 | <0.1 |
| 3 | PdL¹$_R$(OTf) (0.25) | RuL²$_S$ (0.25) | 24 | >99 | 98.8 | 1.0 | 0.2 | <0.1 |
| 4 | PdL¹$_R$(OTf) (2) | RuL²$_R$ (2) | 6 | 37 | <0.1 | 0.1 | 14.8 | 85.0 |
| 5 | PdL¹$_R$(OTf) (2) | RuL²$_R$(2) | 24 | >99 | <0.1 | 0.2 | 14.8 | 84.8 |
| 6 | PdL¹$_R$(OTf) (4) | RuL²$_R$(2) | 24 | >99 | <0.1 | 0.1 | 21.4 | 78.5 |
| 7 | PdL¹$_R$(OTf) (1) | RuL²$_R$(2) | 72 | >99 | <0.1 | 0.2 | 7.1 | 92.7 |
| 8[a] | PdL¹$_R$(OTf) (1) | RuL²$_R$(2) | 72 | 98 | <0.1 | 0.1 | 3.9 | 96.0 |
| 9[a] | PdL³$_R$(OTf) (1) | RuL²$_R$(2) | 72 | 36 | <0.1 | 0.2 | 8.9 | 90.9 |
| 10[a] | PdL⁴$_R$(OTf) (1) | RuL²$_R$(2) | 72 | <5 | n.d. | n.d. | n.d. | n.d. |
| 11[a] | PdL⁵$_R$(OTf) (1) | RuL²$_R$(2) | 72 | 91 | <0.1 | 0.1 | 3.7 | 96.1 |
| 12 | PdL¹$_R$(OTs) (1) | RuL²$_R$(2) | 72 | <5 | n.d. | n.d. | n.d. | n.d. |
| 13 | PdL¹$_R$(BF₄) (1) | RuL²$_R$(2) | 72 | 38 | <0.1 | 0.1 | 12.0 | 87.9 |
| 14[a] | NiL¹$_R$(OTf) (1) | RuL²$_R$(2) | 72 | <5 | n.d. | n.d. | n.d. | n.d. |
| 15[a] | CuL⁶$_R$(OTf) (1) | RuL²$_R$(2) | 72 | <5 | n.d. | n.d. | n.d. | n.d. |
| 16[a] | CuL²$_R$(OTf) (1) | RuL²$_R$(2) | 72 | <5 | n.d. | n.d. | n.d. | n.d. |
| 17 | TfOH (2) | RuL²$_R$ (2) | 12 | <5 | n.d. | n.d. | n.d. | n.d. |
| 18 | p-TsOH (2) | RuL²$_R$ (2) | 12 | <5 | n.d. | n.d. | n.d. | n.d. |
| 19 | PdL¹$_R$(OTf) (2) | (0) | 12 | 0 | n.d. | n.d. | n.d. | n.d. |

[a][**1a**] = 125 mM, [**2a**] = 150 mM.

1 mol% of each catalyst; the results are illustrated in Table 2. A methyl group connecting to the carbonyl group in **1a** could be replaced with H, Et, *i*Pr, and methoxymethyl group (entries 1, 2, 3, 4, and 6, respectively) while maintaining a high level of regio-, diastereo-, or enantioselectivities. The sterically hindered *t*Bu group completely suppressed the reactivity (entry 5). The phenyl group was an acceptable substituent (entry 7). Furthermore, the phenyl group could be functionalized with electron-donating or -withdrawing groups (entries 8 and 9). The α-methylated keto ester afforded the product with a quaternary carbon stereogenic center (entry 10). The results for cyclic substrate were also acceptable (entry 13). The α-fluorinated substrate gave a corresponding product containing tertiary alkyl fluoride (entry 12); however, the α-benzylated substrate did not exhibit any reactivity (entry 11). Next, the generality of allylic alcohol was examined using β-keto ester **1a**. Fluoro, methyl, methoxy, chloro, and trifluoromethyl groups could be introduced into the phenyl group of cinnamyl alcohol at ortho, meta, or para positions (entries 14–24). The phenyl group could be replaced with other aromatic rings such as 2-naphthyl, *N*-Boc-3-pyrrolyl, and *N*-Boc-3-indolyl groups (entries 25–27). Allylic alcohols connecting to the aliphatic substituent gave a linear product as a major product, while enantioselectivity at the α-position was low (64:36

enantiomeric ratio (er); entry 28). This is the first example wherein an aliphatic allyl alcohol gives an allylated product in high yield by using a Ru–L²$_S$ system; this is in contrast to the previously reported system, which gave 1,3-diene as the major product[37]. Concerning stereochemistry, in the standard reaction, the *R,S*-configured *syn*-**3aa** was obtained in the PdL¹$_R$(OTf)/RuL²$_S$ system. For other products (entries 2, 9, 14, 17, 20, and 27), the same absolute configuration was afforded as the major stereoisomers (further details are provided in the Supplementary Information).

Table 3 shows the generality in the mismatched-catalyst system, PdL¹$_R$(OTf) and RuL²$_R$. In β-keto ester, ethyl- and phenyl-substituted keto esters demonstrated similar results to that of the standard substrate in a matched system; however, the *i*Pr-type substrate was not applied (entry 1–4). α-Fluoro β-keto ester could also be used (entry 5). The scope of allylic alcohol was similar to that in the matched system.

## Proposed mechanism

Figure 3 illustrates the proposed catalytic cycles. The matched (blue) and mismatched (red) systems proceeded via similar mechanisms (Fig. 3a). In the blue-colored system, PdL¹$_R$(OTf) reacts with substrate **1** to form the Pd enolate (PdL¹$_R$EN), wherein the enolate anionic ligand

## Table 2 | Generality of the PdL$^1_R$(OTf)/RuL$^2_S$-matched catalyst system[a]

| Entry | Product 3 | Yield (%) | b/l | syn/anti | er[b] |
|---|---|---|---|---|---|
| 1[c,d] | R = H | 82 | >95:5 | >95:5 | >99:1 (R,S) |
| 2 | R = Me | 99 | >95:5 | >95:5 | >99:1 (R,S) |
| 3[e] | R = Et | 98 | >95:5 | >95:5 | >99:1 |
| 4[e] | R = iPr | 91 | >95:5 | >95:5 | >99:1 |
| 5[f] | R = tBu | <5 | n.d. | n.d. | n.d. |
| 6[e,f] | R = CH$_2$OCH$_3$ | 96 | >95:5 | >95:5 | >99:1 |
| 7[e] | R = H | 97 | >95:5 | >95:5 | 99:1 |
| 8[e] | R = p-CH$_3$O | 95 | >95:5 | >95:5 | >99:1 |
| 9 | R = p-Cl | 92 | >95:5 | >95:5 | >99:1 (R,S) |
| 10[e,g,h] | R = Me | 91 | >95:5 | >95:5 | >99:1 |
| 11[i] | R = Bn | <5 | n.d | n.d | n.d |
| 12[e,g,h] | R = F | 89 | >95:5 | >95:5 | 99:1 |
| 13[e,g,h] | | 97 | >95:5 | >95:5 | >99:1 |
| 14 | R = o-F | 99 | >95:5 | >95:5 | >99:1 (R,S) |
| 15[e] | R = o-Me | 97 | >95:5 | >95:5 | 99:1 |
| 16[e] | R = o-CH$_3$O | 96 | >95:5 | >95:5 | >99:1 |
| 17 | R = m-F | 99 | >95:5 | >95:5 | >99:1 (R,S) |
| 18[e] | R = m-Me | 98 | >95:5 | >95:5 | >99:1 |
| 19[e] | R = m-CH$_3$O | 96 | >95:5 | >95:5 | >99:1 |
| 20 | R = p-F | 99 | >95:5 | >95:5 | 99:1 (R,S) |
| 21[e] | R = p-Cl | 97 | >95:5 | >95:5 | >99:1 |
| 22[e,h,j] | R = p-CF$_3$ | 87 | 93:7 | >95:5 | 99:1 |
| 23[e] | R = p-Me | 98 | >95:5 | >95:5 | 99:1 |
| 24[e] | R = p-CH$_3$O | 99 | >95:5 | >95:5 | >99:1 |
| 25 | | 95 | >95:5 | >95:5 | >99:1 (R,S) |
| 26[e,g] | | 97 | >95:5 | >95:5 | >99:1 |
| 27[h] | | 93 | >95:5 | >95:5 | >99:1 (R,S) |

## Table 2 (continued)

1 + 2 → 1 mol% PdL$^1_R$(OTf) / 1 mol% RuL$^2_S$ / 1,4-dioxane / 10 °C → syn-3 + anti-3

| Entry | Product 3 | Yield (%) | b/l | syn/anti | er[b] |
|---|---|---|---|---|---|
| 28 | (structure with Ph, CO$_2$tBu) | 48 | <5:95 | — | 64:36 |

[a]Conditions unless otherwise specified: [**1**] = 500 mM; [**2**] = 600 mM; [PdL$^1_R$(OTf)] = [RuL$^2_S$] = 5.00 mM; 1,4-dioxane 1.00 mL; 10 °C; 24 h.

[b]Er; enantiomer ratio. Symbols in parentheses are absolute configuration of major stereoisomer. Details on determination are provided in the Supplementary Information.

[c][**1**] = 500 mM; [**2**] = 750 mM; [PdL$^1_R$(OTf)] = 1.25 mM; [RuL$^2_S$] = 2.50 mM.

[d]Product was isolated as a β-hydroxy ester after one pot-reduction using K-selectride.

[e]Relative and absolute configurations were not determined. Structures drawn were estimated from the structure-confirmed analogs.

[f]Reaction time 48 h.

[g][**1**] = 250 mM; [**2**] = 500 mM; [PdL$^1_R$(OTf)] = [RuL$^2_S$] = 2.50 mM.

[h]Reaction time 72 h.

[i]4 mol% of PdL$^1_R$(OTf) and 4 mol% of RuL$^2_S$ were used.

[j]2 mol% of PdL$^1_R$(OTf) and 4 mol% of RuL$^2_S$ were used.

## Table 3 | Generality of the PdL$^1_R$(OTf)/RuL$^2_R$-mismatched catalyst system.[a]

1 + 2 → 1 mol% PdL$^1_R$(OTf) / 2 mol% RuL$^2_R$ / 1,4-dioxane / 10 °C → syn-3 + anti-3

| Entry | Product 3 | | Yield (%) | b/l | syn/anti | er[b] |
|---|---|---|---|---|---|---|
| 1 | | R = Me | 99 | >95:5 | <5:95 | >99:1 (R,R) |
| 2[c] | | R = Et | 94 | >95:5 | 8:92 | >99:1 |
| 3 | (structure with Ph, CO$_2$tBu) | R = iPr | <5 | n.d. | n.d. | n.d. |
| 4[c,d] | | R = Ph | 93 | >95:5 | <5:95 | >99:1 |
| 5[c,d,e] | (structure with F, Ph, CO$_2$tBu) | | 92 | >95:5 | 6:94 | >99:1 |
| 6[c] | | R = o-Me | 92 | >95:5 | 5:95 | 99:1 |
| 7 | | R = m-F | 95 | >95:5 | 5:95 | >99:1 |
| 8 | (structure with aryl-R, CO$_2$tBu) | R = p-F | 91 | >95:5 | 6:94 | >99:1 |
| 9[c] | | R = p-Cl | 95 | >95:5 | <5:95 | >99:1 |
| 10[c] | | R = p-Me | 97 | >95:5 | 5:95 | >99:1 |

[a]Conditions unless otherwise specified: [**1**] = 125 mM, [**2**] = 150 mM; [PdL$^1_R$(OTf)] = 1.25 mM; [RuL$^2_R$] = 2.50 mM; 1,4-dioxane 4.00 mL; 10 °C; 72 h.

[b]Er; enantiomer ratio. Symbols in parentheses are absolute configuration of major stereoisomer. The details on determination are provided in the Supplementary Information.

[c]Relative and absolute configurations were not determined. Structures drawn were estimated from structure-confirmed analogs.

[d]96 h.

[e][PdL$^1_R$(OTf)] = [RuL$^2_R$] = 2.50 mM.

coordinates to the Pd metal in a bidentate manner. In addition, the reaction liberates TfOH, which collaborates with RuL$^2_S$ to activate the allylic alcohol based on the RDACat mechanism[36] to form endo-π-allyl species (RuL$^2_S$All), releasing a water molecule. The Pd enolate then attacks the allyl group, producing adduct **3** with the regeneration of PdL$^1_R$(OTf) and RuL$^2_S$. In the π-allyl complex RuL$^2_S$All formation, one of the two possible diastereomers was selectively generated owing to chiral ligand L$^2_S$. The substituent R$^2$ on the π-allyl group is placed at one of the syn-positions away from the sterically hindered dioxolane

moiety of the chiral ligand L$^2_S$; thus, ruthenium occupies the Si face on C(3) of the π-allyl ligand (when R$^2$ is Ph) as illustrated for RuL$^2_S$All[37]. In PdL$^1_R$EN, the two pseudo-equatorial phenyl groups on the phosphinyl group of BINAP shield the planal enolate, and the tBu group on the carboxylate stands up to avoid steric repulsion with a phenyl group of BINAP. Consequently, the two phenyl groups and a tBu group occupy the spaces, which is similar to that illustrated for PdL$^1_R$EN[41]. The Pd enolate approaches the RuL$^2_S$All from outside to generate (R,S)-**3** as the major isomer. The enantioface selections are consistent with those

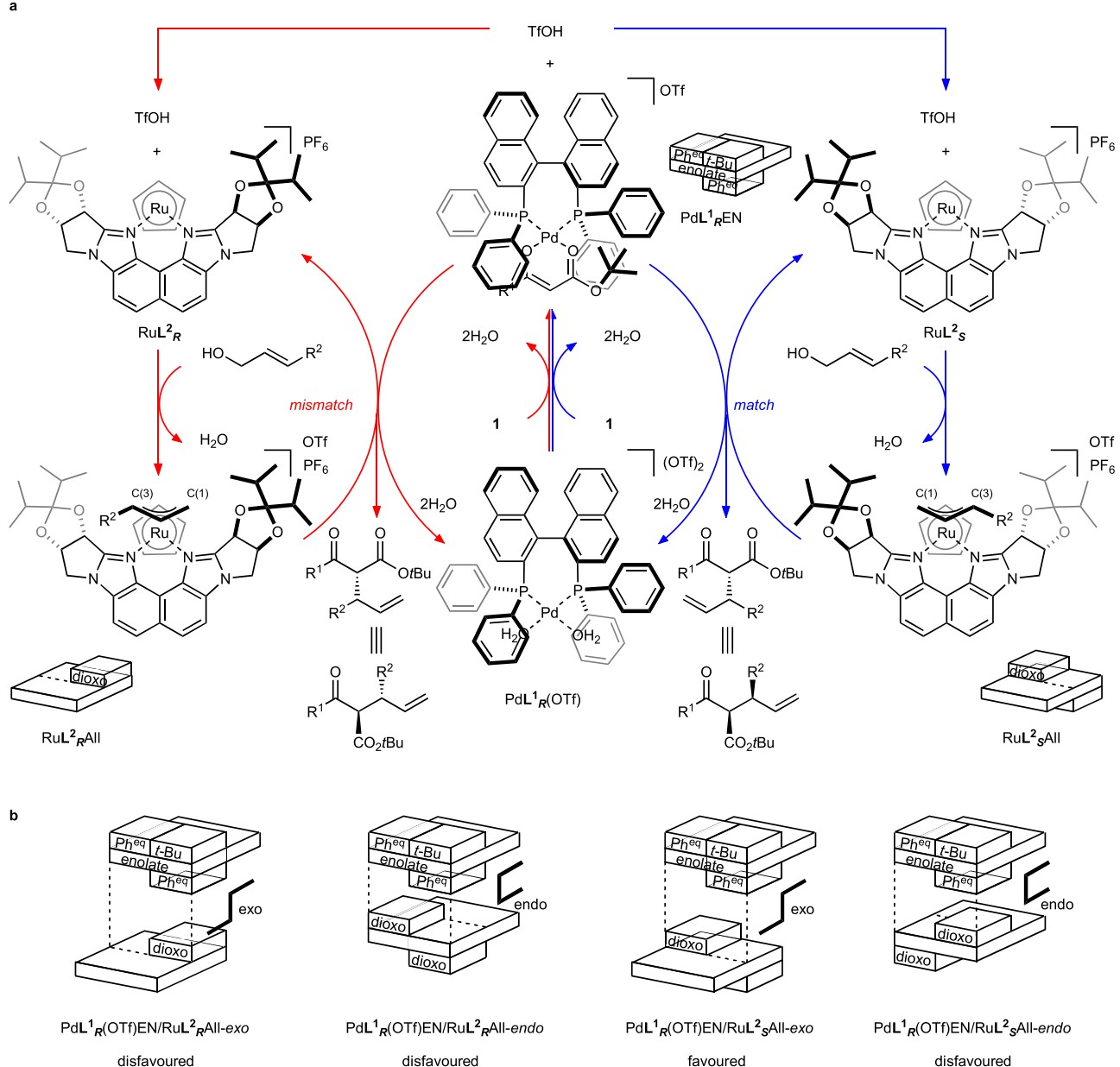

**Fig. 3 | Proposed reaction mechanism. a** Catalytic cycle of the matched (blue) and mismatched (red) systems using PdL$^1_R$(OTf) complex with RuL$^2_S$ or RuL$^2_R$ complex. **b** Pattern diagram of nucleophilic addition for realizing the origin of diastereoselectivity. EN enolate ligand, All allyl ligand.

in the previous reports[37,40]. The bulkiness of the *t*Bu group on the ester is important, and also demonstrated in the previous report[40]. Moreover, a methyl ester substrate corresponding to **1a** gave ca. 1:1 diastereomeric mixture. In both matched and mismatched cases, the enantioface selectivity of the π-allyl moiety was almost perfect, and the selection of Pd enolate was highly affected by the steric repulsion.

Models for the nucleophilic attack in the matched or mismatched system are shown in Fig. 3b. The nucleophilic attack proceeds via one of the two patterns by considering the relative stereochemistry between the binaphthyl backbone of BINAP (**L**$^1$) and the naphthalene backbone of Naph-diPIM-dioxo-*i*Pr (**L**$^2$) at exo and endo positions. The PdL$^1_R$EN/RuL$^2_S$All-*exo* intermediate had good stereocomplementarity in the matched system, whereas the endo structure exhibited steric repulsion. A similar mechanism was followed in the mismatched system. Although the endo intermediate was better than the exo intermediate, it was unfavorable as the backbones closed each other. Furthermore, the mismatched system

had low reactivity and diastereoselectivity due to the steric repulsion of both the intermediates.

Over allylation product was not obtained in the current catalytic system. The phenomenon is similar to a PdL$^1$(OTf)-catalyzed α-alkylation of α-alkyl β-keto esters, in which the acceptable substrates are limited to only smaller substituents[40]. Enolate formation of the allylated β-keto ester with PdL$^1$(OTf) is presumably difficult (Table 2, entry 11) owing to steric reasons. The nucleophilic addition does not proceed even when the enolate is formed and the enolate may be protonated stereoselectively to reverse the mono-allylated product **3**[42,43].

### Diastereoselective reduction of carbonyl group
The reduction of the carbonyl group of the adduct **3** gave a βHE, generating another stereogenic center. Three adjacent chiral carbon centers can be controlled by employing diastereoselective reduction, which is summarized in Fig. 4. For example, (*R*,*S*)-*syn*-**3aa**, the product

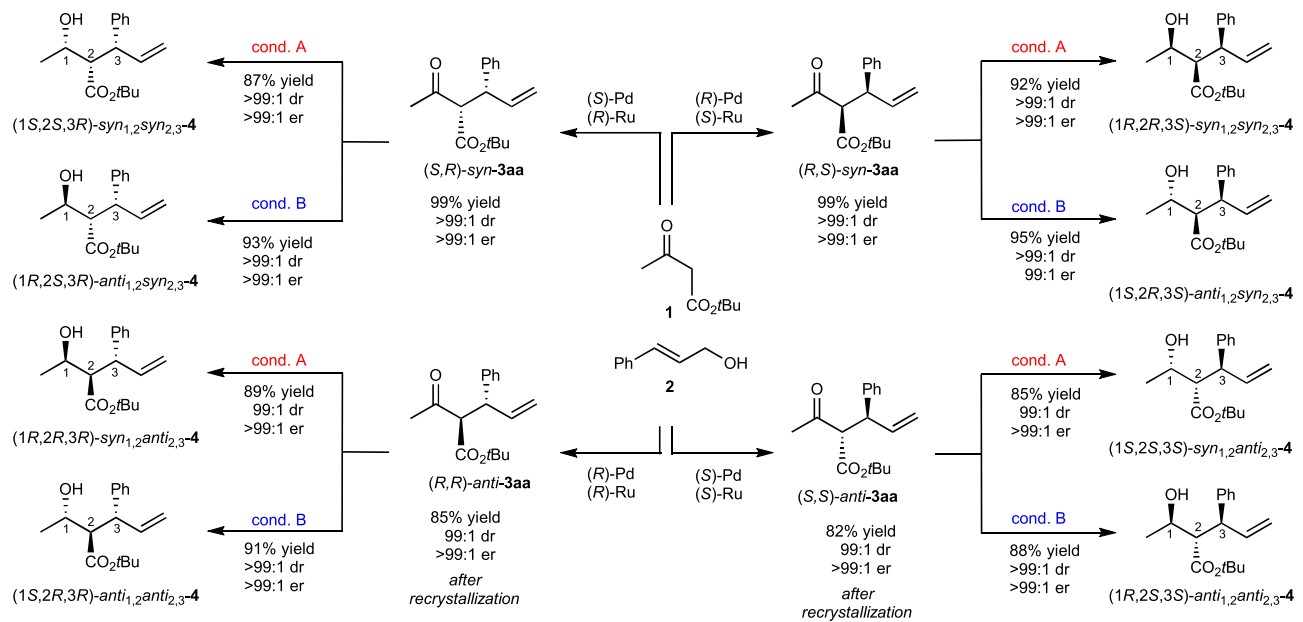

**Fig. 4 | Acyclic stereocontrolled synthesis of eight possible isomers with three adjacent stereogenic centers.** Condition A: [**3aa**] = 0.1 M, K-selectride 0.35 M, THF, −78 °C, 12 h; Condition B: [**3aa**] = 0.1 M, Zn(BH₄)₂ 0.1 M, CH₂Cl₂-Et₂O, 0 °C, 6 h. THF: tetrahydrofuran.

generated in the matched Pd**L¹**$_R$(OTf)/Ru**L²**$_S$ system, was reduced by K-selectride (cond. A) to afford *syn,syn*-**4**. The origin of diastereoselectivity is further explained by the Felkin−Ahn model. The Luche reduction using LaCl₃/NaBH₄ also gave *syn,syn*-**4** in high selectivity (93:7 dr). In contrast, the substrate (*R,S*)-*syn*-**3aa** was reduced to another diastereomer, *anti,syn*-**4**, using Zn(BH₄)₂ (cond. B)[44]. In this case, the reduction proceeded via the Zimmerman−Traxler chelation model. Enantiomeric (*S,S,R*)-*syn,syn*-**4** and (*R,S,R*)-*anti,syn*-**4** were obtained from (*S,R*)-*syn*-**3aa**. Diastereomeric adducts **3**s, (*R,R*)- and (*S,S*)-*anti*-substrates, could also be reduced with high diastereoselectivity by the same methods. A total of eight diastereomers were synthesized by introducing these reduction methods combined with stereodivergent allylation.

## Formal synthesis of (+)-pancratistatin

Thus, compounds with highly condensed stereogenic centers can be synthesized using the above method. These compounds can be further extended to synthesize complex molecules owing to the easily transformative carboxylic ester and olefinic moiety. Herein, we applied our method to the formal synthesis of (+)-pancratistatin to demonstrate the utility of the products[45–47].

The result is illustrated in Fig. 5 (further details are provided in the Supplementary Information). The β-keto ester **1n** and the allylic alcohol **2q** were dehydratively condensed using the above catalytic method (0.5 mol% of Pd**L¹**$_R$(OTf) and Ru**L²**$_S$, 1,4-dioxane, 25 °C, 48 h, >99% yield (NMR), >99:1 er, 99:1 dr). The product was reduced in the one-pot reaction under Luche's conditions to afford the hydroxy ester **5** as a single diastereomer in 92% total yield after the isolation process. The internal double bond was converted to oxirane via hydroxy group-triggered regio-/diastereoselective epoxidation by VO(acac)₂/*tert*-butyl hydroperoxide (TBHP), in 94% yield and 92:8 dr[48]. The epoxide moiety was converted to allylic alcohol via regioselective ring-opening by sequential nucleophilic selenol addition and oxidation of selenyl ether followed by β-elimination[49]. The corresponding product **7** was cyclized to cyclohex-3-en-1,2-diol **8** by ring-closing metathesis using 1 mol% of Grubb's 2nd generation catalyst and obtained in 91% yield. Ester hydrolysis of **8** gave carboxylic acid **9** in quantitative yield. The Curtius rearrangement of acid **9** followed by the CH₃OK treatment gave intermediate **10** in 92% yield; the intermediate was previously

reported by Hudlicky et al.[46]. Although the number of steps and the total yield were not the most outstanding[47], the chemical yield of each step was high. Thus, stereocontrolling is simple when each chiral catalyst is appropriately used, and various stereoisomers can be synthesized by changing the chirality of the catalysts and the types of reagents. Additionally, the usage of protecting groups can be reduced.

In summary, we established enantio- and diastereoselective dehydrative allylation of β-keto esters in a stereodivergent manner using a binary catalyst system involving Pd**L¹**(OTf) and Ru**L²**. This method can be applied to α-non-substituted keto esters to generate products with tertiary stereogenic centers, which are easily epimerized to a diastereomeric mixture under acidic or basic conditions. During the dual catalytic cycle, Pd**L¹**(OTf) and keto ester substrates generate Pd-enolate species and liberate a Brønsted acid; the acid acts as a co-catalyst of Ru**L²** to form π-allyl species and water. This concerted mechanism facilitates nearly neutral conditions to avoid product epimerization. Although the Pd**L¹**(OTf)/Ru**L²** combination exhibited matched/mismatched stereocomplimentarity, both combinations gave >95:5 diastereoselectivity. A wide range of generality allowed the changing of the substituents at the β-position of the keto ester, and aromatic rings of allylic alcohol were realized. Moreover, a quaternary stereogenic carbon center was constructed using α-alkyl β-keto ester. By introducing diastereoselective reduction of the carbonyl group, a total of eight possible diastereomeric isomers could be synthesized on demand. The utility of the highly-stereogenic-center-condensed product was realized by the formal synthesis of (+)-pancratistatin. With the increasing importance of stereodivergent synthesis owing to recent achievement[50–53], this method can further improve the utility of beneficial acetoacetic ester synthesis.

## Method
### General procedure for dehydrative allylation of β-keto ester
The detailed operation was described by taking the dehydrative allylation of β-keto ester **1a** with allylic alcohol **2a** catalyzed by Pd**L¹**$_R$(OTf) and Ru**L²**$_S$ (matched system) as the representative (Table 2, entry 2). A 10 mL Young-type Schlenk tube was charged with [RuCp(CH₃CN)₃]PF₆ (2.16 mg, 4.97 μmol), **L²**$_S$ (2.73 mg, 5.01 μmol), and CH₂Cl₂ (1.0 mL). The mixture was stirred at rt for 30 min, and then the resulting pale-yellow solution was concentrated in vacuo. To these were added Pd**L¹**$_R$(OTf)

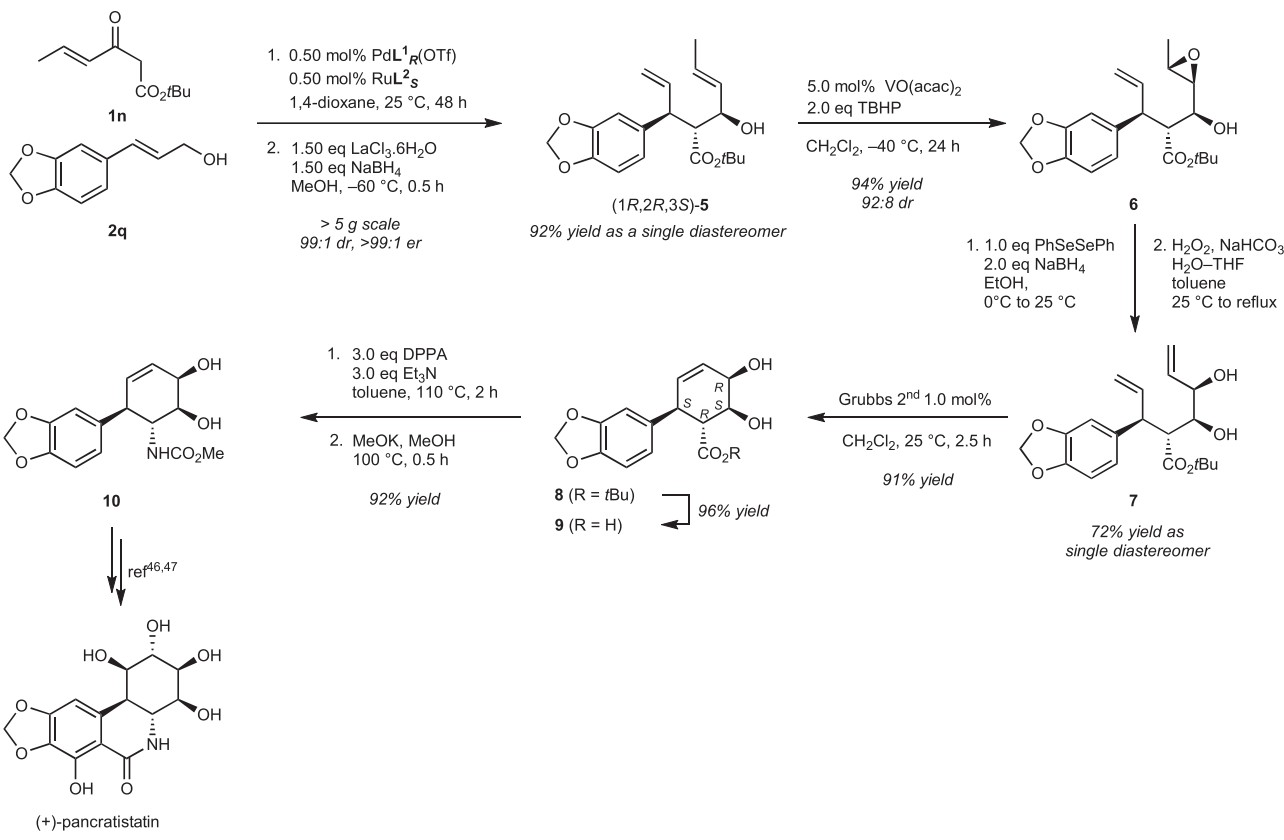

**Fig. 5 | Formal synthesis of (+)-pancratistatin.** TBHP *tert*-butyl hydroperoxide, THF tetrahydrofuran, DPPA diphenylphosphoryl azide.

(5.30 mg, 4.99 µmol), **1a** (81.5 µL, 79.0 mg, 0.500 mmol) and a 0.600 M 1,4-dioxane solution of **2a** (1.00 mL, 0.600 mmol) under Ar atmosphere. The resulting red-brown solution was stirred at 10 °C for 12 h, resulting in a pale-yellow solution. The reaction mixture was filtered through a short-pad of silica gel and washed two times by a 10:1 hexane/EtOAc mixture. The filtrate was concentrated and purified by using neutralized SiO$_2$-chromatography (20 g; toluene eluent, note: to prevent epimerization of product, flash chromatography was performed as fast as possible, normally within 5 min) to produce *syn*-**3aa** as a white solid (136.9 mg, 99.8%).

## Data availability
All crystallographic data generated in this study have been deposited in the Cambridge Crystallographic Data Center (CCDC) database under accession code 2193030 ((R,S)-*syn*-**3aa**), 2193039 ((R,R)-*anti*-**3aa**), 2193045 ((1R,2R,3S)-*syn*$_{1,2}$*syn*$_{2,3}$-**11aa**), 2193050 ((1R,2R,3R)-*syn*$_{1,2}$*anti*$_{2,3}$-**11aa**), 2193051 ((1S,2R,3R)-*anti*$_{1,2}$*anti*$_{2,3}$-**11aa**), 2193052 ((2R,3S)-*syn*-**11ba**), 2193053 ((R,S)-*syn*-**3ia**), 2193054 ((1R,2R,3S)-*syn*$_{1,2}$-*syn*$_{2,3}$-**11ab**), 2193059 ((1R,2R,3S)-*syn*$_{1,2}$*syn*$_{2,3}$-**11ae**), 2193062 ((1R,2R,3S)-*syn*$_{1,2}$*syn*$_{2,3}$-**11ah**), 2193063 ((1R,2R,3S)-*syn*$_{1,2}$*syn*$_{2,3}$-**11am**), 2193064 ((R,S)-*syn*-**3ao**), and 2193066 (**8**). For compound numbers, see Supplementary Information. All other data relating to the findings of this study, NMR spectrum, and HPLC charts are provided in the Supplementary Information.

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

## Acknowledgements
This work was supported by JSPS KAKENHI (Grant Numbers JP16H02274 (MK), JP20K21191 (MK), and JP21K05052 (ST)), the Advanced Catalytic Transformation Program for Carbon Utilization (ACT-C, JPMJCR12YC) (MK), and ERATO (JPMJER2103) from the Japan Science and Technology Agency (JST) (ST), and The Naito Foundation (ST). LPT is thankful to the Kobayashi Foundation for the Special Research Fellowship.

## Author contributions
S.T. and M.K. designed this study. L.P.T., M.Y., and S.T. conducted the experiments. M.K. and K.S. oversaw the project. S.T., K.S., and M.K. wrote the manuscript. L.P.T., M.Y., and S.T. wrote the Supplementary Information.

## Competing interests
The authors declare no competing interests.
