## [Peer Review File · Nature Communications]

REVIEWER COMMENTS

Reviewer #1 (Remarks to the Author):

In this manuscript, Prof. Tanaka and Prof. Kitamura and their co-workers report a stereodivergent allylation of β -keto esters with allylic alcohols. This transformation is enabled by a synergistic Pd/Ru system. Here the cationic BinapPd(II)(OTf)₂ plays the role of Lewis acid to activate the β -keto esters to form the enolate under neutral conditions and at the same time, catalytic TfOH is generated. The TfOH enters the Ru catalytic cycle to facilitate the dehydrative formation of π -allyl-Ru from allylic alcohols. Then the asymmetric allylation between the nucleophiles (Pd-enolate) and electrophiles (π -allyl-Ru) afford the product. Stereodivergent fashion is achieved because the Pd catalyst and Ru catalyst independently control the stereochemistry of the nucleophile and electrophile. Therefore, all four stereoisomers of the products could be selectively obtained by tuning the chirality of the two metal catalysts. Notably, by merging with a thereafter stereodivergent ketone reduction (reagent controlled), synthesis of eight possible isomers with three adjacent stereogenic centers is demonstrated. To my knowledge, these are very impressive results in stereodivergent synthesis. Finally, the authors have applied this method to a concise synthesis of (+)-pancratistatin, further increasing the bright spot for this work. Stereodivergent synthesis involving β -keto esters is still an unsolved challenge. This is because the β -keto esters are easy to occur in ketone-enol isomerization and therefore strong single background reaction without a second catalyst to activate the nucleophiles would happen. As a result, it is difficult to pursue synergistic catalysis for the stabilized nucleophiles (β -keto esters belong to typical stabilized nucleophiles). The critical success of this Pd/Ru system is that it occurs in a very neutral condition (without the addition of acid or base); therefore, the background single catalysis is prohibited. I like this manuscript very much and strongly recommend its publication in Nature communication!

Some minor revisions might be considered to improve the manuscript.

1. H⁺ is not an exogenously added catalyst (it comes from BinapPd(II)(OTf)₂ catalyst activating β -keto esters and the deprotonate by the OTf⁻). So "Using a Ru/H⁺/Pd Synergistic Catalyst" in the title is inaccurate. I suggest revising it to "Using Synergistic Ru/Pd Catalysis"
2. The sentence "The phenyl group could also be substituted with the keto group (entry 7)" is difficult to understand. Please consider revising it.
3. "RDACat mechanism" is a too professional term; please avoid using the abbreviation.

4. Based on the stereochemistry rationalization (Fig. 3), the bulk tBu-group of β -keto esters is crucial to obtaining the stereocontrol, (similar results are also observed in previously reported work of Pd-enolate formation, as shown in ref 35, 36, and 39). However, could the authors offer some results in the manuscript about methyl β -keto esters and ethyl β -keto esters. These results could give the reader a better understanding of stereocontrol mode.

5. tBu Ketone (R1 = tBu) (entry 5, Table 2) is not workable for this system. Especially for the mismatched catalyst combination, R1= iPr is not working (entry 3, Table 3). I guess this is caused by the steric hindrance. I think that, in such a case, is it possible to use a less steric ester moiety (CO₂Me, for example) to improve the reactivity? The R1 = tBu might also provide enough steric to obtain stereo controlling, but maybe a reverse absolute configuration might give for the nucleophile part? I think it deserves to check these interesting substrates.

6. Could a little more substrate scope be given for the mismatched system in Table 3? For example, the keto-esters generating quaternary stereocenters can be provided (like the one shown in entries 7 to 9 in Table 2)

7. The stereochemistry rationalization part is a little difficult to understand. Did these rationalizations have solid support for them? As we know that π -allyl-M could occur σ - π - σ isomerization and the Ru catalyst used herein is not C₂-symmetric. So there should be four stereoisomeric π -allyl-Ru in each system (no matter the matched or mismatched being discussed). These conditions were not fully considered in the discussions. Would the Cp group lead to the reference of some stereoisomeric π -allyl-Ru? Should there be an equilibrium between those stereoisomeric π -allyl-Ru, the thereafter C-C bond formation is likely a dynamic kinetic transformation process? In this case, the C-C bond formation is both enantio- and diastereodetermining step.

8. If in the current stage, the stereochemistry could not be well-explained, I suggest the authors delete this part and just give a rough mechanism and let the stereochemistry studies (this part is very complex because two stereocenters are produced and the stereodivergent process is given, especially in a synergistic catalyst system) to be detailed investigated in the future works. I believe this work is still an excellent contribution to Nature Communication even without the Fig. 3 part.

Weiwei Zi from Nankai University

Reviewer #2 (Remarks to the Author):

Stereodivergent synthesis realized by bimetallic catalysis has emerged as a powerful and attractive strategy in accessing to all the stereoisomers of a molecule. In this study, the author used Pd catalyst to activate the pronucleophile β -keto esters and Ru catalyst to activate allylic alcohol, thus the reaction could undergo at mild reaction conditions and provide the products in very high yields. Also, by coordinating the two metal catalysts with chiral ligands, all four stereoisomers of the product were yielded, and this is a significant advantage of bimetallic catalysis. Furthermore, the author successfully avoided using base in the reaction, so that the racemic problem of the products was solved. In addition, the further transformations of the product realize the construction of the third stereogenic center and the eight stereoisomers are afforded. The formal synthesis of (+)-pancratistatin also prove the pragmatic potential of this reaction. In conclusion, I believe that this article can be published only after the following problems and questions are adequately addressed:

1) In the title, the statement “a Ru/H⁺/Pd Synergistic Catalyst” is not appropriate enough. Although the proton from the pronucleophile engages in the activation of the allylic alcohol, the three parts, Ru catalyst, Pd catalyst, and proton, do not work synergistically. Thus, I believe that “a Ru/Pd Synergistic Catalyst” is much more reasonable description. Secondly, the statement “Stereocontrol of Three Adjacent Stereogenic Centers” is also not warranted, since the third stereogenic center is formed in the subsequent reduction of the product and stereodivergent dehydrative allylation only constructs two stereogenic centers.

2) In Fig. 1 all the serial numbers of the group (R) should be superscript.

3) In Table 3, entry 3, the yield should be < 5%

4) Usually, due to the instrumental error, the b/l and syn/anti can not be recorded as high as >99:1, but just >20:1.

5) There are some updated works on bimetallic catalyzed stereodivergent synthesis that should be referred: *CCS Chemistry*, 2022, 4, 1720-1731; *Angew. Chem. Int. Ed.* 2021, 60, 24941-24949; *J. Am. Chem. Soc.* 2021, 143, 12622-12632; *Chem* 2022, DOI: 10.1016/j.chempr.2022.04.006.

6) This reaction can give branched product with excellent regioselectivity for the aryl substituted allylic alcohol, but it can only give linear product when the allylic alcohol is alkyl substituted. The explanation should be added.

7) In general, bimetallic catalytic system mainly focuses on rare precious metal/rare precious metal and rare precious metal/earth-abundant metal. The Pd catalyst performs as a Lewis acid in this reaction, and have you investigated much more Lewis acids that are less expensive than Pd ?

8) Only tert-butyl esters are used in this reaction, what about other esters?

REPLYS TO REVIEWERS' COMMENTS

Reviewer #1 (Remarks to the Author):

Dear Professor Weiwei Zi from Nankai University;

In this manuscript, Prof. Tanaka and Prof. Kitamura and their co-workers report a stereodivergent allylation of β -keto esters with allylic alcohols. This transformation is enabled by a synergistic Pd/Ru system. Here the cationic BinapPd(II)(OTf)₂ plays the role of Lewis acid to activate the β -keto esters to form the enolate under neutral conditions and at the same time, catalytic TfOH is generated. The TfOH enters the Ru catalytic cycle to facilitate the dehydrative formation of π -allyl-Ru from allylic alcohols. Then the asymmetric allylation between the nucleophiles (Pd-enolate) and electrophiles (π -allyl-Ru) afford the product. Stereodivergent fashion is achieved because the Pd catalyst and Ru catalyst independently control the stereochemistry of the nucleophile and electrophile. Therefore, all four stereoisomers of the products could be selectively obtained by tuning the chirality of the two metal catalysts. Notably, by merging with a thereafter stereodivergent ketone reduction (reagent controlled), synthesis of eight possible isomers with three adjacent stereogenic centers is demonstrated. To my knowledge, these are very impressive results in stereodivergent synthesis. Finally, the authors have applied this method to a concise synthesis of (+)-pancratistatin, further increasing the bright spot for this work. Stereodivergent synthesis involving β -keto esters is still an unsolved challenge. This is because the β -keto esters are easy to occur in ketone-enol isomerization and therefore strong single background reaction without a second catalyst to activate the nucleophiles would happen. As a result, it is difficult to pursue synergistic catalysis for the stabilized nucleophiles (β -keto esters belong to typical stabilized nucleophiles). The critical success of this Pd/Ru system is that it occurs in a very neutral condition (without the addition of acid or base); therefore, the background single catalysis is prohibited. I like this manuscript very much and strongly recommend its publication in Nature communication!

Reply: We highly appreciate Reviewer 1 for his positive evaluation. We are also grateful that he read our manuscript deeply, understood our chemistry, and gave valuable comments.

Some minor revisions might be considered to improve the manuscript.

Reply: Thank you again for important comments. Reply comments and modified points are given point by point below.

1. H^+ is not an exogenously added catalyst (it comes from $\text{BinapPd(II)(OTf)}_2$ catalyst activating β -keto esters and the deprotonate by the OTf-). So "Using a Ru/ H^+ /Pd Synergistic Catalyst" in the title is inaccurate. I suggest revising it to "Using Synergistic Ru/Pd Catalysis"

Reply: We agree with his comment. We do not utilize any acids in this system, and that is the important point to show high diastereoselectivity. Thus, H^+ has been removed from the title.

2. The sentence "The phenyl group could also be substituted with the keto group (entry 7)" is difficult to understand. Please consider revising it.

Reply: This sentence was changed to "The phenyl group was an acceptable substituent (entry 7)" to make it simple and understandable. This part is highlighted in the revised manuscript.

3. "RDACat mechanism" is a too professional term; please avoid using the abbreviation.

Reply: This abbreviation which appears firstly was changed to "Redox-mediated Donor-Acceptor Bifunctional Catalyst (RDACat)". This part is highlighted in the revised manuscript.

4. Based on the stereochemistry rationalization (Fig. 3), the bulk *t*Bu-group of β -keto esters is crucial to obtaining the stereocontrol, (similar results are also observed in previously reported work of Pd-enolate formation, as shown in ref 35, 36, and 39). However, could the authors offer some results in the manuscript about methyl β -keto esters and ethyl β -keto esters. These results could give the reader a better understanding of stereocontrol mode.

Reply: As the Reviewer pointed out, the bulky *t*Bu group at ester moiety is essential to show high diastereoselectivity, which is a similar observation reported in the previous report (ref 40 in the revised manuscript). When the methyl ester was adopted in our catalytic system, low diastereoselectivity was observed. These descriptions, "The bulkiness of the *t*Bu group on the ester is important, and also demonstrated in the previous report.⁴⁰ Moreover, a methyl ester substrate corresponding to **1a** gave ca. 1:1 diastereomeric mixture." was added to the mechanism section. This part is highlighted in the revised manuscript.

5. *t*Bu Ketone ($R_1 = t\text{Bu}$) (entry 5, Table 2) is not workable for this system. Especially for the mismatched catalyst combination, $R_1 = i\text{Pr}$ is not working (entry 3, Table 3). I guess this is caused by the steric hindrance. I think that, in such a case, is it possible to use a less steric

ester moiety (CO₂Me, for example) to improve the reactivity? The R1 = *t*Bu might also provide enough steric to obtain stereo controlling, but maybe a reverse absolute configuration might give for the nucleophile part? I think it deserves to check these interesting substrates.

Reply: We agree that the low reactivities of these substrates are due to a steric problem. The Reviewer's suggestion is very interesting, and the β -keto ester having a neopentyl group (CH₂*t*Bu), which is the mimics of OtBu moiety, may be a good substrate to show high selectivity with COOMe (see also ref 40 in the revised manuscript). This opinion is highly suggestive for us to expand the substrate scope of the reaction.

6. Could a little more substrate scope be given for the mismatched system in Table 3? For example, the keto-esters generating quaternary stereocenters can be provided (like the one shown in entries 7 to 9 in Table 2)

Reply: In a mismatched system, we have examined a trend of the reactivity and selectivity to compare with the matched system. As the result, the reactivities are generally lower. We have examined the keto-ester substrate to generate quaternary stereocenter, unfortunately, the reaction proceeded a little. Since the main purpose of this paper is construction of tertiary chiral carbon center, we did not examine the reaction further.

7. The stereochemistry rationalization part is a little difficult to understand. Did these rationalizations have solid support for them? As we know that π -allyl-M could occur σ - π - σ isomerization and the Ru catalyst used herein is not C₂-symmetric. So there should be four stereoisomeric π -allyl-Ru in each system (no matter the matched or mismatched being discussed). These conditions were not fully considered in the discussions. Would the Cp group lead to the reference of some stereoisomeric π -allyl-Ru? Should there be an equilibrium between those stereoisomeric π -allyl-Ru, the thereafter C-C bond formation is likely a dynamic kinetic transformation process? In this case, the C-C bond formation is both enantio- and diastereodetermining step.

Reply: As the Reviewer pointed out, usually, the π -allyl-ruthenium complex can isomerize via π - σ - π equilibrium. However, due to the steric effect of chiral ligand L²s, a single diastereomer is selectively generated in our system. This would be the origin of high enantioselectivity. The structure of the π -allyl complex is already confirmed in the previous report by X-ray crystallographic analysis (see ref 37 in the revised manuscript). We agree that

a discussion about this stereochemistry is not fully described. Thus, we have added following sentences to the mechanism section. "In the *endo*- π -allyl complex formation, one of the two possible diastereomers was selectively generated owing to chiral ligand L^2_S ." "The Pd enolate approaches the RuL^2_SAlI from outside to generate (*R,S*)-**3** as the major isomer. The enantioface selections are consistent with those in the previous reports.^{37,40}" These parts are highlighted in the revised manuscript. Thank you for an important comment. Because a C_2 -symmetrical chiral ligand L^2_S is utilized, the configuration of the Cp group and L^2_S on the ruthenium does not affect the stereochemistry, and only two diastereomers are possible in the *endo*- π -allyl complex.

8. *If in the current stage, the stereochemistry could not be well-explained, I suggest the authors delete this part and just give a rough mechanism and let the stereochemistry studies (this part is very complex because two stereocenters are produced and the stereodivergent process is given, especially in a synergistic catalyst system) to be detailed investigated in the future works. I believe this work is still an excellent contribution to Nature Communication even without the Fig. 3 part.*

Reply: Thank you for your valuable suggestion. We agree with the Reviewer's opinion. The current catalytic cycle is complex due to the synergistic system based on two chiral catalysts. Further, we do not have any definite proof so far. However, enantioface selection mechanisms of each chiral catalyst are already well discussed in previous papers. Since the stereo-selection manners are identical, we can apply them to our catalysis. Thus, a core part of the mechanism is a discussion of the difference between match and mismatch systems. To make it more obvious, the description of match/mismatch stereocomplementarity was separated as an independent paragraph. This part is highlighted in the revised manuscript.

Reviewer #2 (Remarks to the Author):

Stereodivergent synthesis realized by bimetallic catalysis has emerged as a powerful and attractive strategy in accessing to all the stereoisomers of a molecule. In this study, the author used Pd catalyst to activate the pronucleophile β -keto esters and Ru catalyst to activate allylic alcohol, thus the reaction could undergo at mild reaction conditions and provide the products in very high yields. Also, by coordinating the two metal catalysts with chiral ligands, all four

stereoisomers of the product were yielded, and this is a significant advantage of bimetallic catalysis. Furthermore, the author successfully avoided using base in the reaction, so that the racemic problem of the products was solved. In addition, the further transformations of the product realize the construction of the third stereogenic center and the eight stereoisomers are afforded. The formal synthesis of (+)-pancratistatin also prove the pragmatic potential of this reaction. In conclusion, I believe that this article can be published only after the following problems and questions are adequately addressed:

Reply: We also highly appreciate Reviewer 2. His/Her comments are also very suggestive. Reply comments and modified points are given point by point below.

1) In the title, the statement “a Ru/H+/Pd Synergistic Catalyst” is not appropriate enough. Although the proton from the pronucleophile engages in the activation of the allylic alcohol, the three parts, Ru catalyst, Pd catalyst, and proton, do not work synergistically. Thus, I believe that “a Ru/Pd Synergistic Catalyst” is much more reasonable description. Secondly, the statement “Stereocontrol of Three Adjacent Stereogenic Centers” is also not warranted, since the third stereogenic center is formed in the subsequent reduction of the product and stereodivergent dehydrative allylation only constructs two stereogenic centers.

Reply: The first part of this comment is the same suggestion to comment 1 of Reviewer 1, please refer to the reply to it. In the second part, as the Reviewer’s comment, this description may lead to a misunderstanding that the stereodivergent allylation can control the three adjacent stereogenic centers. Thus, we have removed them from the title.

2) In Fig. 1 all the serial numbers of the group (R) should be superscript.

Reply: These were modified to superscripts. Thank you for your careful annotation.

3) In Table 3, entry 3, the yield should be < 5%

Reply: This part was also modified.

4) Usually, due to the instrumental error, the b/l and syn/anti can not be recorded as high as >99:1, but just >20:1.

Reply: According to the Reviewer’s suggestion, the values are changed to >95:5 in which the ratio is higher than 96:4.

5) *There are some updated works on bimetallic catalyzed stereodivergent synthesis that should be referred: CCS Chemistry, 2022, 4, 1720-1731; Angew. Chem. Int. Ed. 2021, 60, 24941-24949; J. Am. Chem. Soc. 2021, 143, 12622-12632; Chem 2022, DOI: 10.1016/j.chempr.2022.04.006.*

Reply: These are added into references 32–35. Thank you for giving the valuable informations.

6) *This reaction can give branched product with excellent regioselectivity for the aryl substituted allylic alcohol, but it can only give linear product when the allylic alcohol is alkyl substituted. The explanation should be added.*

Reply: The origin of the high regioselectivity may be caused by steric and electronic reasons. Because of a steric repulsion of the phenyl group against to ruthenium complex, the C(3) of the π -allyl moiety is away from the ruthenium. Additionally, phenyl substituent stabilizes the carbocation at C(3). Consequently, the electrophilicity of C(3) becomes higher than that of C(1), and nucleophiles attack C(3) selectively. This mechanism was described in a previous report (see ref 37 in the revised manuscript). On the other hand, this is the first experience that alkyl substituted allylic alcohol gave the allylated product in high yield. In previous reports, undesired dehydrated 1,3-diene formed as a major product. In that case, even once the π -allyl-ruthenium(IV) complex forms, reductive γ -proton elimination to form 1,3-diene and ruthenium(II) may occur before the nucleophilic attack. We are also quite interesting and presume that this result is due to the neutral reaction condition. However, unfortunately, enantioselectivity was low. Although we have no evidence for the origin of regioselectivity so far, this would extend to developing another allylation chemistry. Thank you for pointing out the important result. The following sentence was added to the manuscript. "This is the first example wherein an aliphatic allyl alcohol gives an allylated product in high yield by using a Ru–L²S system; this is in contract to the previously reported system, which gave 1,3-diene as the major product.³⁷" This part is highlighted in the revised manuscript. Together with this, "C(1)" and "C(3)" were added to the π -allyl ligand in Figure 3.

7) *In general, bimetallic catalytic system mainly focuses on rare precious metal/rare precious metal and rare precious metal/earth-abundant metal. The Pd catalyst performs as a Lewis acid in this reaction, and have you investigated much more Lewis acids that are less expensive than Pd ?*

Reply: I agree with this opinion that use of the precious metal should be avoided. We have examined other plausible catalysts, Cu, Ni, or H⁺ system. However, unfortunately, no reaction occurred. This result demonstrates the high utility of the Pd complex, which can form an enolate complex with liberating an acid even under neutral conditions. These results are included in Table 1.

8) *Only tert-butyl esters are used in this reaction, what about other esters?*

Reply: Please refer to a reply to comment 4 of Reviewer 1.

ADDITIONAL CHANGES

- 1) According to the journal format, abstract has been modified together with many other modifications. These changes were recorded as track changes.
- 2) The revised manuscript was reviewed by proofreading service, and some parts were modified. These changes were also recorded as track changes.
- 3) Crystallographic data were further refined, and the structure and its data were also changed. Accordingly, flack parameters for each structure, figures for crystal structure (Figures S04, S09, S28, S42, S53, S64, S81, S88, S95, S123, S127, S134, and S141) and tables for crystal structural data (Tables S01–S12 and S14) were renewed. These changes do not affect any result of this study.

REVIEWERS' COMMENTS

Reviewer #1 (Remarks to the Author):

Thank you for the response. All my concerns have been appropriately addressed, and I am satisfied with the author's revisions to the manuscript. Congratulations to the authors for such a good piece of work on stereodivergent synthesis!

On the last thing, considering the rapidly growing reports on stereodivergent synthesis, I suggest the authors consider citing some updated literature on this field.

1. Diastereodivergent Aldol-Type Coupling of Alkoxyallenes with Pentafluorophenyl Esters Enabled by Synergistic Palladium/Chiral Lewis Base Catalysis.

Angew. Chem. Int. Ed. 2022,10.1002/anie.202207621

2. Cooperative Pd/Cu-catalyzed diastereodivergent coupling of allenamides and aldimine esters to access the Mannich-type motifs

Chem. Catal. 2022, 2, 1428-1439.

3. Synergistic Pd/Amine-Catalyzed Stereodivergent Hydroalkylation of 1,3-Dienes with Aldehydes: Reaction Development, Mechanism, and Stereochemical Origins

J. Am. Chem. Soc. 2021, 143, 10948-10962.

4. Diastereodivergent Synthesis of β -Amino Alcohols through Dual-Metal-Catalyzed Coupling of Alkoxyallenes with Aldimine Esters

Angew. Chem. Int. Ed. 2021, 60, 6545-6552.

Reviewer #2 (Remarks to the Author):

After careful revision by the authors, this manuscript has now become a high-quality research paper which is deserved for its publication in Nat. Commun.

REPLYS TO REVIEWERS' COMMENTS

Reviewer #1 (Remarks to the Author):

Dear Professor Weiwei Zi from Nankai University;

Thank you for the response. All my concerns have been appropriately addressed, and I am satisfied with the author's revisions to the manuscript. Congratulations to the authors for such a good piece of work on stereodivergent synthesis!

Reply: We highly appreciate Reviewer 1 for his highly positive evaluation.

On the last thing, considering the rapidly growing reports on stereodivergent synthesis, I suggest the authors consider citing some updated literature on this field.

1. Diastereodivergent Aldol-Type Coupling of Alkoxyallenes with Pentafluorophenyl Esters Enabled by Synergistic Palladium/Chiral Lewis Base Catalysis.

Angew. Chem. Int. Ed. 2022, 10.1002/anie.202207621

2. Cooperative Pd/Cu-catalyzed diastereodivergent coupling of allenamides and aldimine esters to access the Mannich-type motifs

Chem. Catal. 2022, 2, 1428-1439.

3. Synergistic Pd/Amine-Catalyzed Stereodivergent Hydroalkylation of 1,3-Dienes with Aldehydes: Reaction Development, Mechanism, and Stereochemical Origins

J. Am. Chem. Soc. 2021, 143, 10948-10962.

4. Diastereodivergent Synthesis of β -Amino Alcohols through Dual-Metal-Catalyzed Coupling of Alkoxyallenes with Aldimine Esters

Angew. Chem. Int. Ed. 2021, 60, 6545-6552.

Reply: These references were cited with revision of the final sentence. Thank you for suggestions.

Reviewer #2 (Remarks to the Author):

After careful revision by the authors, this manuscript has now become a high-quality research paper which is deserved for its publication in Nat. Commun.

Reply: We are happy to hear that the Reviewer satisfied this revision. Thank you again for your evaluation and valuable suggestions.

ADDITIONAL CHANGES

1) Other changes are highlighted in the revised manuscript.